# Strategies of Advanced Airway Management in Out-of-Hospital Cardiac Arrest during Intra-Arrest Hypothermia: Insights from the PRINCESS Trial

**DOI:** 10.3390/jcm11216370

**Published:** 2022-10-28

**Authors:** Jonathan Tjerkaski, Thomas Hermansson, Emelie Dillenbeck, Fabio Silvio Taccone, Anatolij Truhlar, Sune Forsberg, Jacob Hollenberg, Mattias Ringh, Martin Jonsson, Leif Svensson, Per Nordberg

**Affiliations:** 1Department of Clinical Science and Education, Södersjukhuset, Center for Resuscitation Science, Karolinska Institute, 11883 Stockholm, Sweden; 2Department of Intensive Care, Hôpital Universitaire de Bruxelles (HUB), Université Libre de Bruxelles (ULB), 1070 Brussels, Belgium; 3Emergency Medical Services of the Hradec Kralove Region, 500 05 Hradec Kralove, Czech Republic; 4Department of Anaesthesiology and Intensive Care Medicine, Charles University in Prague, University Hospital Hradec Kralove, 500 05 Hradec Kralove, Czech Republic; 5Department of Medicine, Karolinska Institute, 17176 Solna, Sweden; 6Function Perioperative Medicine and Intensive Care, Karolinska University Hospital, 17176 Stockholm, Sweden

**Keywords:** cardiac arrest, intra-arrest hypothermia, airway management

## Abstract

Background: Trans-nasal evaporative cooling is an effective method to induce intra-arrest therapeutic hypothermia in out-of-hospital cardiac arrest (OHCA). The use of supraglottic airway devices (SGA) instead of endotracheal intubation may enable shorter time intervals to induce cooling. We aimed to study the outcomes in OHCA patients receiving endotracheal intubation (ETI) or a SGA during intra-arrest trans-nasal evaporative cooling. Methods: This is a pre-specified sub-study of the PRINCESS trial (NCT01400373) that included witnessed OHCA patients randomized during resuscitation to trans-nasal intra-arrest cooling vs. standard care followed by temperature control at 33 °C for 24 h. For this study, patients randomized to intra-arrest cooling were stratified according to the use of ETI vs. SGA prior to the induction of cooling. SGA was placed by paramedics in the first-tier ambulance or by physicians or anesthetic nurses in the second tier while ETI was performed only after the arrival of the second tier. Propensity score matching was used to adjust for differences at the baseline between the two groups. The primary outcome was survival with good neurological outcome, defined as cerebral performance category (CPC) 1–2 at 90 days. Secondary outcomes included time to place airway, overall survival at 90 days, survival with complete neurologic recovery (CPC 1) at 90 days and sustained return of spontaneous circulation (ROSC). Results: Of the 343 patients randomized to the intervention arm (median age 64 years, 24% were women), 328 received intra-arrest cooling and had data on the airway method (*n* = 259 with ETI vs. *n* = 69 with SGA). Median time from the arrival of the first-tier ambulance to successful airway management was 8 min for ETI performed by second tier and 4 min for SGA performed by the first or second tier (*p* = 0.001). No significant differences in the probability of good neurological outcome (OR 1.43, 95% CI 0.64–3.01), overall survival (OR 1.26, 95% CI 0.57–2.55), full neurological recovery (OR 1.17, 95% CI 0.52–2.73) or sustained ROSC (OR 0.88, 95% CI 0.50–1.52) were observed between ETI and SGA. Conclusions: Among the OHCA patients treated with trans-nasal evaporative intra-arrest cooling, the use of SGA was associated with a significantly shorter time to airway management and with similar outcomes compared to ETI.

## 1. Introduction

Out-of-hospital cardiac arrest (OHCA) is a major public health concern, affecting approximately 275,000 individuals each year in Europe [1]. The overall OHCA mortality rate is approximately 90%, with lifelong disabilities being common among the survivors [2]. Airway management and ventilation is an important element of the advanced cardiac life support (ACLS) protocol, which has been formulated in order to improve outcomes for OHCA victims [3]. Currently, most patients receive advanced airway management during resuscitation, either using endotracheal intubation (ETI) or a supraglottic airway (SGA) [4].

ETI has been used by emergency medical services (EMS) since the 1970s [5]. However, several studies have questioned the safety and effectiveness of ETI performed by EMS in the pre-hospital setting [6,7]. The potential harms of pre-hospital ETI include unrecognized tube misplacement or dislodgement, iatrogenic hyperventilation and chest compression interruptions during placement [8,9,10]. SGA insertion is most often simpler and faster to insert than ETI [11], which results in higher success rates and fewer interruptions in the administration of chest compressions [12,13,14]. Despite the supposed benefits of SGA, several observational studies have suggested that ETI may be associated with better outcomes than SGA [15]. However, recent randomized controlled trials have raised some controversies on this issue [13,14]. Thus, the optimal strategy for airway management in OHCA remains unclear.

Targeted temperature management remains an important intervention that may influence survival with good neurological function among cardiac arrest patients [16]. In particular, intra-arrest cooling using trans-nasal evaporative cooling may provide some benefits on neurologic recovery in patients with initial shockable rhythms (i.e., ventricular fibrillation or pulseless ventricular tachycardia) [17,18,19]. Trans-nasal evaporative intra-arrest cooling has emerged as a promising therapeutic strategy in OHCA [18,20,21]. The use of SGA in these cases has the potential to shorten the time to successful airway management and thereby enable a shorter time to start cooling. However, as SGA may be associated with an increased risk for aspiration, it is important to examine the safety of advanced airway management in the setting of trans-nasal evaporative intra-arrest cooling therapy.

In this sub-study of the PRINCESS trial, we aimed to compare the effect on neurologic outcome among OHCA patients that had received airway management with ETI versus SGA prior to trans-nasal evaporative intra-arrest cooling.

## 2. Methods

### 2.1. Study Design

We performed a post hoc sub-analysis of data from the PRINCESS trial, which is a multicenter randomized clinical trial that compared trans-nasal evaporative intra-arrest cooling to the standard ACLS in a bystander-witnessed OHCA (Trial registration: NCT01400373) [20]. Ethics and institutional committees in each of the participating countries approved the study protocol [22]. Written informed consent was obtained from the next of kin or a legal representative after hospital admission and from each study participant who regained mental capacity. In this sub-analysis, the patients in the intervention arm in PRINCESS were the primary study population that was subsequently divided into two groups depending on the strategy for airway management. 

### 2.2. Study Participants

We included bystander witnessed OHCA randomized to the intervention arm in the PRINCESS trial. The exclusion criteria were age ≥80 years; an etiology of cardiac arrest due to trauma, head trauma, severe bleeding, drug overdose, cerebrovascular accident, drowning, smoke inhalation, electrocution, or hanging; hypothermia at the time of evaluation; an anatomical barrier preventing the insertion of intra-nasal catheters; an existing do-not-attempt resuscitation order; known terminal illness; known or clinically apparent pregnancy; known coagulopathy (except when therapeutically induced); need for supplemental oxygen; ROSC prior to randomization; and EMS response time (i.e., from collapse to EMS arrival) greater than 15 min. In this sub-study of the PRINCESS trial, we also excluded study participants who were randomized to the control group as well as study participants who were initially randomized to the intervention group but did not receive intra-arrest cooling. Patients were divided into two different treatment groups depending on the airway management technique used prior to trans-nasal evaporative cooling (i.e., those receiving ETI versus those receiving SGA).

### 2.3. Emergency Medical Services

All study sites had two-tiered EMS systems where the first vehicle used bag mask ventilation only or SGA with bag-valve ventilation connected to the SGA prior to the arrival of the second tier. The second tier was manned by physicians or anesthetic nurses, trained in advance airway management including placing an SGA and endotracheal intubation. In addition, the intra-arrest cooling equipment was carried by the second tier. Thus, patients could have had the SGA placed by paramedics from the first vehicle and subsequently, cooling was started after randomization and application of trans-nasal evaporative cooling by the crew in the second tier. Among patients receiving bag mask ventilation by the paramedics, ETI was performed after the arrival of the second tier. The use of SGA could also be due to ETI being difficult to perform in the field. No data were collected on the number of intubation attempts or change in airway strategy. The confirmation of the tube was undertaken with end tidal CO_2_, but this was only recorded in a limited number of patients and not presented in this analysis. 

### 2.4. Exposure

The exposure of interest was defined as the type of airway management technique used prior to cooling. All of the study participants included in the PRINCESS trial were treated with advanced airway management prior to randomization using either ETI or an SGA [22]; patients were therefore divided into two different treatment groups depending on the airway management technique used prior to trans-nasal evaporative cooling.

### 2.5. Treatment

The RhinoChill™ device delivers a mixture of air or oxygen and a chemically inert cooling liquid (perfluorohexane) via nasal catheters directly into the nasal cavity, with the goal of primarily cooling the brain [20,21,23]. Trans-nasal evaporative cooling is maintained until hospital arrival, and whenever possible until systemic cooling is initiated. The study participants received standard post-resuscitation care upon admission to the intensive care unit (ICU). Intravenous sedation, analgesia, and neuromuscular blockade were used according to the institutional cooling protocols. The targets for respiratory management, blood pressure, and glucose control have been previously described [22]. The study participants were treated with targeted temperature management at 32–34 °C for 24 h. 

### 2.6. Outcome

Neurological outcome assessment was performed at 90 days via a structured telephone interview or during a follow-up appointment using the cerebral performance categories (CPC) scale [24]. The primary outcome of this study was survival with good neurologic outcome (CPC 1–2) at 90 days. The secondary outcomes were overall survival at 90 days, survival with complete neurologic recovery (CPC 1) at 90 days, and hospital admission with the sustained return of spontaneous circulation (ROSC) (defined as ROSC >20 min). Additional safety parameters that were investigated included the time until successful airway management, the time until the initiation of intra-arrest cooling, arterial blood gas parameters, and the prevalence of pneumonia. The time until successful airway management was defined as the time interval that elapsed between EMS arrival at the site of the arrest and the time of successful airway device placement. Similarly, we defined the time until the initiation of trans-nasal evaporative cooling as the time duration between EMS arrival and the initiation of intra-arrest cooling.

### 2.7. Statistical Analysis

Continuous variables were presented as means and standard deviations (SD) if normally distributed, or as medians and interquartile ranges (IQR) if not normally distributed. Categorical variables were reported as counts and percentages. We assessed the group differences in continuous variables using either the Mann–Whitney U-test or Student’s *T*-test, as appropriate. Group-wise differences in categorical variables were assessed using Pearson’s chi-squared test.

A range of factors may influence the decision of EMS personnel to use one advanced airway device or strategy over any other form of airway management. Therefore, we used propensity score matching to balance known confounding variables across the two treatment groups, in a manner reminiscent of that conducted in the work of Hasegawa et al. and McMullan et al. [4,7]. Propensity scores were calculated using a logistic regression model with the following independent variables: EMS response time, age, sex, bystander CPR, initial rhythm, etiology, and body mass index (BMI). Propensity score matching (1:2) was carried out using the nearest neighbor method with a caliper width of 0.2. We used the standardized mean difference (SMD) to examine group differences in the covariates before and after matching. We fit a series of conditional logistic regression models to evaluate the association between each outcome variable and airway management strategy. We calculated bootstrapped 95% confidence intervals for the odds ratios using 1000 bootstrapped datasets. 

We used multiple imputation by chained equations (mice) to impute missing data, generating five imputed datasets [25]. The analysis described above was separately performed in each of the imputed datasets. The resulting regression coefficients and test statistics were subsequently pooled across all imputed datasets [25,26].

## 3. Results

### 3.1. Study Population

Of the 343 patients who were allocated to intra-arrest cooling in the PRINCESS trial, six did not receive the assigned intervention, and data on airway management were missing for nine study participants (Figure 1). Thus, the study cohort consisted of 328 patients; 259 (79%) received orotracheal intubation (ETI group), while in the SGA group, six were treated using a laryngeal tube and the others (*n* = 63) with a laryngeal mask airway. Patients who were treated with SGA were older (*p* = 0.03) and had a higher BMI (*p* = 0.01) than patients who were treated with ETI (Table 1). After propensity score matching, all baseline characteristics were adequately balanced between groups (Appendix A). No significant differences in the arterial blood gas parameters at the time of hospital admission were observed between groups after propensity score matching (Table 2). Missing entries amounted to 3.32% of the included data (Appendix A).

### 3.2. Outcome Measures

No significant differences between ETI and SGA were observed on the occurrence of survival with good neurologic outcome, CPC 1-2 at 90 days (OR 1.43, 95% CI 0.64–3.01), overall survival at 90 days (OR 1.26, 95% CI 0.57–2.55), survival with complete neurologic recovery at 90 days (OR 1.17, 95% CI 0.52–2.73) or hospital admission following sustained ROSC (OR 0.88, 95% CI 0.50–1.52), as can be seen in Figure 2. 

### 3.3. Time until Successful Airway Device Placement and the Initiation of Intra-Arrest Cooling

The time until airway management was available for 276 out of 328 study participants (84%). The median time from EMS arrival at the scene until successful airway management was 8 (interquartile range 4–12 min) minutes in the ETI group and 4 min in the SGA group (interquartile range 2–7 min), *p* = 0.001 (Figure 3A). However, we did not find any statistically significant difference between the SGA- and ETI groups regarding the time until the initiation of hypothermia treatment, which was on average 14 min in the ETI group (interquartile range 8–20) and 15 min in the SGA group (interquartile range 11–25), *p* = 0.053 (Figure 3B). 

The time until the initiation of intra-arrest cooling therapy was associated with survival with good neurological outcome (defined as CPC 1-2, OR 0.96 [0.92–1.00], *p* = 0.045), overall survival (OR 0.95 [0.91–0.99], *p* = 0.017), and survival with complete neurologic recovery at 90 days (defined as CPC 1, 0.95 [0.90–0.99], *p* = 0.018), as shown in the Appendix A. In contrast, we did not observe any statistically significant association between the likelihood of achieving sustained ROSC and the time of intra-arrest cooling. Moreover, we found no statistically significant association between the endpoints of this study and the time until successful airway device placement. Thus, whereas the time until the initiation of intra-arrest cooling has a statistically significant association with both neurological outcomes and overall survival in OHCA patients, the time until successful airway device placement does not have any significant relationship to patient outcomes. 

### 3.4. Infections

The incidence of infections prior to hospital discharge was reported in conjunction with the PRINCESS trial by all participating centers (Figure 4). Approximately 20% of patients suffered from some form of infection during their hospital stay, with pneumonia accounting for the majority of these cases. We found no statistically significant differences between the ETI- and SGA groups in the proportion of patients who suffered from pneumonia during their hospital stay (*p* = 0.579).

## 4. Discussion

In this sub-analysis of the PRINCESS trial, we observed that SGA is feasible to use without any safety aspects reported prior to induction of trans-nasal evaporative cooling in OHCA and it significantly shortened the airway management time compared to ETI; SGA was not associated with any worsening in the gas exchange on arterial blood gas or with patient outcomes.

Trans-nasal evaporative intra-arrest cooling is an emerging therapeutic option in the management of OHCA patients which has, as the only cooling method, been shown to be safe and effective in inducing intra-arrest cooling at the scene of the arrest. Although not currently part of routine medical practice, trans-nasal evaporative intra-arrest cooling therapy has already been implemented in some clinical settings for the treatment of OHCA. Furthermore, trans-nasal evaporative intra-arrest cooling will continue to be investigated in clinical trials. Therefore, it is important to establish the safety of advanced airway management in the setting of trans-nasal evaporative intra-arrest cooling. In this study, we observed a shorter time period to successful airway management in the SGA group compared to intubation, but no differences in terms of sustained ROSC, overall survival, neurological recovery, the prevalence of pneumonia, or arterial blood gas parameters.

We observed that patients who were treated with SGA were on average older and had a higher BMI than patients who were treated with ETI. Although these imbalances were adjusted for in the subsequent analyses, we cannot dismiss the possibility of residual confounding due to one or several unmeasured parameters such as the lack of equipoise at the baseline including the expertise of centers. Therefore, we believe that there is a need for an external validation to confirm our results.

Despite concerns regarding an increased aspiration risk following trans-nasal evaporative cooling, we did not find any significant differences between the ETI and SGA groups concerning the proportion of patients suffering from pneumonia or arterial blood gas parameters on admission to hospital. Despite a limited sample size, these findings further support our other results, which suggest that ETI and SGA have similar safety profiles in the setting of trans-nasal evaporative intra-arrest cooling.

We observed a statistically significant difference between the ETI and SGA groups in the time elapsed until successful airway placement in the setting of trans-nasal evaporative intra-arrest cooling therapy. This result is in agreement with that of earlier studies on the topic of advanced airway management in OHCA [5,11]. However, we observed no statistically significant differences concerning the time until the initiation of trans-nasal evaporative intra-arrest cooling therapy. This result is most likely attributed to logistics, as the cooling device was carried by the second-tier vehicles and not by the paramedics in the first-tier ambulances. The pre-hospital physicians generally arrived at the site of the arrest later than the paramedics. Thus, although the patient’s airway may have been secured using a SGA by the paramedic prior to the arrival of the pre-hospital emergency medicine physicians, intra-arrest cooling could only be initiated once the physician had arrived at the site of the arrest, making the time to hypothermia largely independent of the time until successful airway management.

This secondary analysis of the PRINCESS trial had several limitations. The study of two subgroups receiving different airway strategies within the intervention arm of the PRINCESS trial introduced a risk of selection bias. Although the treatment groups had similar baseline characteristics and propensity score matching was performed, the risk of residual confounding could not be eliminated. An additional limitation of this study is the fact that ETI and SGA could not be compared to bag-valve mask ventilation, as successful advanced airway management was specified as a requirement for inclusion in the PRINCESS trial [22]. In addition, the time elapsed from EMS arrival and successful ETI including the time interval between the arrival of the first- and second-tier vehicle and time interval for the ETI procedure, which may have enabled the SGA, could be undertaken by the first-tier team, to be placed faster. Furthermore, data on the quality of CPR such as information regarding chest compression interruption were unavailable and could thus not be included in this analysis. We also lacked information on the intubation attempts and where SGA was placed due to intubation failure. 

## 5. Conclusions

In this sub-study of the PRINCESS trial, a shorter time period to successful airway management was observed in the SGA group when compared to ETI. No differences in clinically relevant outcomes such as survival with good neurologic outcome and overall survival were observed between groups. This study might help to design future trials using trans-nasal evaporative cooling to minimize the time to induce cooling.

## Figures and Tables

**Figure 1 jcm-11-06370-f001:**
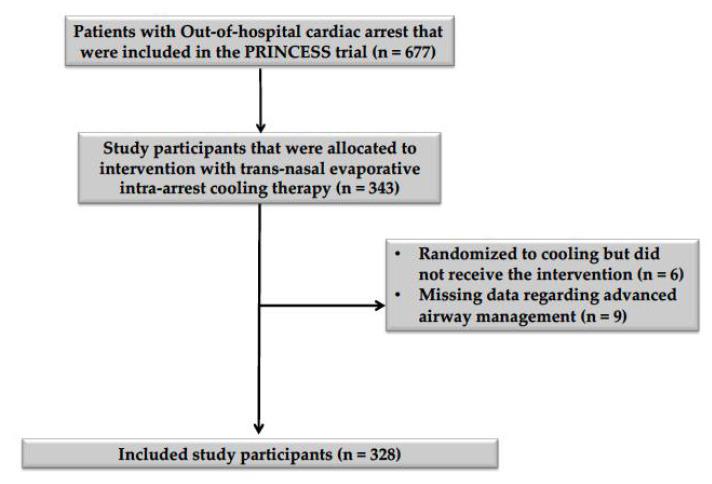
Flowchart describing patient inclusion and exclusion. A diagram describing the inclusion and exclusion of study participants, according to the inclusion and exclusion criteria mentioned in the methodology section of this article.

**Figure 2 jcm-11-06370-f002:**
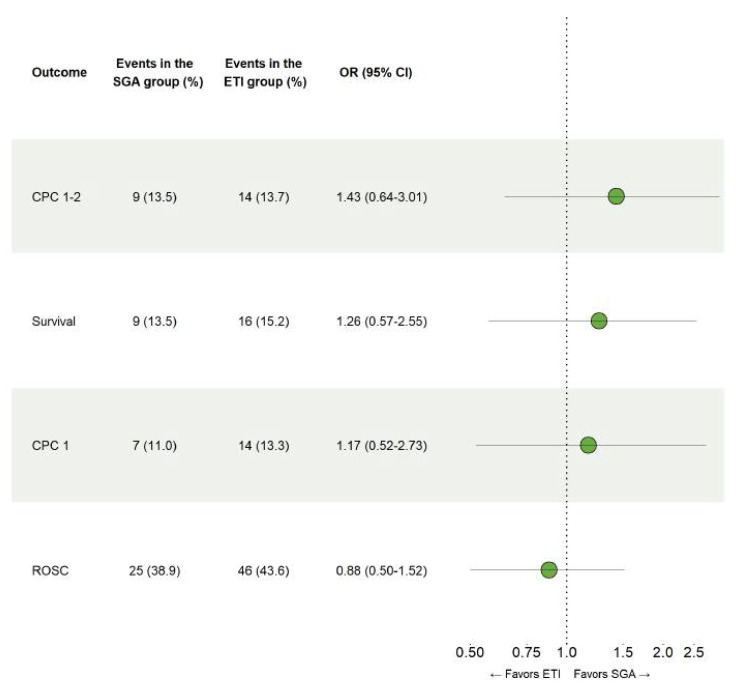
Comparison of endotracheal intubation and supraglottic airways. Results for the conditional logistic regression models that were fitted using propensity score matched data to examine the safety of ETI and SGA in patients with out-of-hospital cardiac arrest.

**Figure 3 jcm-11-06370-f003:**
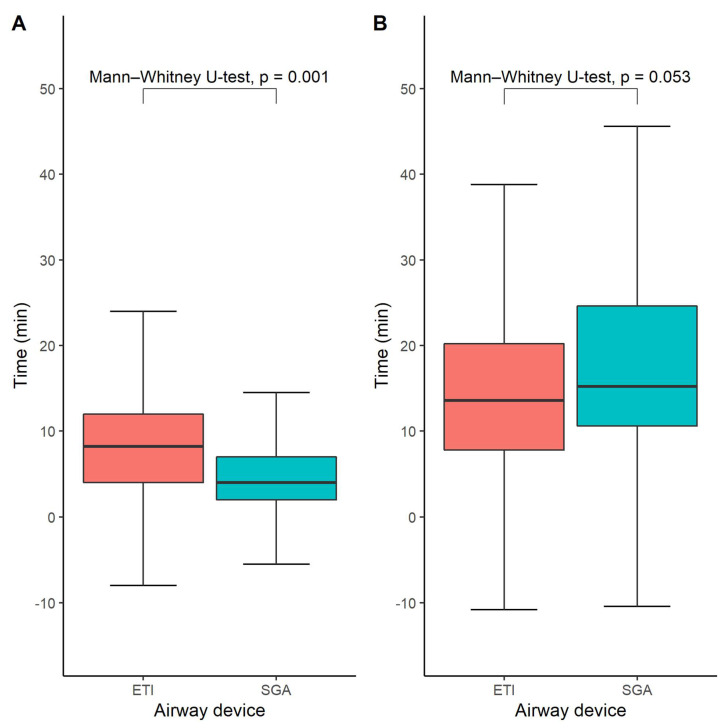
Time until successful advanced airway management (**A**) and time to start of intra-arrest cooling (**B**). Box-and-whisker plots depicting the time until successful airway management in the ETI and SGA groups, respectively. Medians and quartiles were estimated using quantile regression for each imputed dataset and subsequently averaged across all imputed datasets. Likewise, the Mann–Whitney U-test was performed in each imputed dataset and the results were pooled across all available imputed datasets.

**Figure 4 jcm-11-06370-f004:**
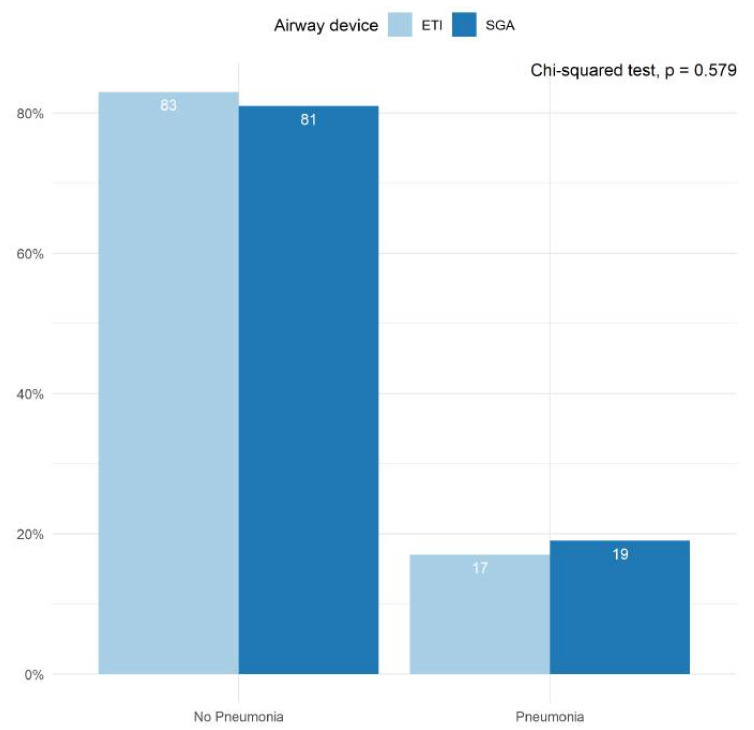
Pneumonia diagnosed during hospital stay. Bar plots depicting the proportion of patients in whom pneumonia was diagnosed prior to hospital discharge. Results are shown for the ETI group and the SGA group separately. The proportions were averaged across all imputed datasets. The Chi-squared test was performed in each imputed dataset and the results were pooled across all available imputed datasets.

**Table 1 jcm-11-06370-t001:** Demographics before and after propensity score matching. Data after propensity score matching corresponded to that of one of the five imputed datasets that were generated following multiple imputation by chained equations (mice). Abbreviations: BMI = body mass index, CPR = cardiopulmonary resuscitation, EMS = emergency medical services, ETI = endotracheal intubation, SD = standard deviation, SGA = supraglottic airway, SMD = standardized mean difference, Q1 = first quartile, Q3 = third quartile.

	Unmatched			Matched		
	ETI (N = 259)	LMA (N = 69)	SMD	ETI (N = 126)	LMA (N = 67)	SMD
Sex						
Female	64 (24.7%)	18 (26.1%)	0.03	28 (22.2%)	18 (26.9%)	0.10
Male	195 (75.3%)	51 (73.9%)	−0.03	98 (77.8%)	49 (73.1%)	−0.10
Location						
At home	126 (55.5%)	46 (70.8%)	0.31	85 (67.5%)	47 (70.1%)	0.06
Other	16 (7.0%)	4 (6.2%)	−0.08	10 (7.9%)	4 (6.0%)	−0.06
Outside	54 (23.8%)	5 (7.7%)	−0.60	9 (7.1%)	5 (7.5%)	0.03
Public place	31 (13.7%)	10 (15.4%)	0.08	22 (17.5%)	11 (16.4%)	−0.06
Etiology						
Other	34 (14.3%)	13 (20.6%)	0.21	27 (21.4%)	15 (22.4%)	0.00
Suspected cardiac	204 (85.7%)	50 (79.4%)	−0.21	99 (78.6%)	52 (77.6%)	0.00
Bystander CPR						
No	89 (35.7%)	23 (35.9%)	0.07	48 (38.1%)	27 (40.3%)	0.03
Yes	160 (64.3%)	41 (64.1%)	−0.07	78 (61.9%)	40 (59.7%)	−0.03
EMS arrival time (minutes)						
Median (Q1–Q3)	7.0 (5.0, 10.0)	7.5 (4.0, 12.0)	0.04	8.0 (5.0, 11.0)	8.0 (4.5, 11.5)	−0.11
Time to ROSC (minutes)						
Median (Q1–Q3)	22.0 (15.0, 37.0)	23.5 (14.8, 29.5)	−0.28	24.0 (15.5, 40.5)	25.0 (17.0, 36.0)	0.11
Hospital admission time (minutes)						
Median (Q1–Q3)	48.0 (37.0, 59.0)	48.0 (39.0, 58.0)	−0.45	46.0 (35.0, 56.5)	51.0 (42.0, 58.8)	0.30
Initial rhythm						
Non-shockable	149 (57.8%)	47 (68.1%)	0.22	84 (66.7%)	45 (67.2%)	0.02
Shockable	109 (42.2%)	22 (31.9%)	−0.22	42 (33.3%)	22 (32.8%)	−0.02
Age (years)						
Mean (SD)	61.6 (12.1)	65.4 (12.3)	0.31	64.8 (10.7)	65.5 (12.2)	0.05
BMI						
Median (Q1–Q3)	26.2 (24.2, 29.3)	27.8 (25.4, 30.9)	0.31	27.8 (24.7, 30.1)	27.8 (25.0, 30.5)	0.02

**Table 2 jcm-11-06370-t002:** Patient characteristics at hospital admission. The presented results correspond to data obtained after propensity score matching for one of the five imputed datasets that were generated following multiple imputation by chained equations (mice). Abbreviations: PaCO_2_ = partial pressure of carbon dioxide, PaO_2_ = partial pressure of oxygen, SD = standard deviation, Q1 = first quartile, Q3 = third quartile.

	ETI	LMA	*p* Value
Lactate (mmol/L)	*n* = 24	*n* = 17	0.491
Mean (SD)	11.1 (4.9)	10.5 (5.3)	
Median (Q1–Q3)	11.2 (9.2, 13.2)	10.0 (7.0, 13.3)	
Range (Min–Max)	3.2–21.0	2.6–19.0	
PaCO_2_ (mmHg)	*n* = 26	*n* = 19	0.491
Mean (SD)	60.6 (25.5)	64.9 (32.2)	
Median (Q1–Q3)	54.0 (38.6, 84.0)	60.8 (46.9, 72.0)	
Range (Min–Max)	33.0–113.2	33.0–178.5	
PaO_2_ (mmHg)	*n* = 26	*n* = 18	0.390
Mean (SD)	86.7 (48.1)	96.7 (46.6)	
Median (Q1–Q3)	74.8 (48.6, 106.3)	86.6 (65.7, 127.3)	
Range (Min–Max)	31.5–202.5	30.3–213.0	
pH	*n* = 24	*n* = 20	0.683
Mean (SD)	7.0 (0.2)	7.1 (0.2)	
Median (Q1–Q3)	7.0 (6.9, 7.2)	7.0 (6.9, 7.2)	
Range (Min–Max)	6.7–7.4	6.7–7.3	
Mean arterial pressure (mmHg)	*n* = 24	*n* = 17	0.233
Mean (SD)	82.9 (24.2)	90.3 (23.9)	
Median (Q1–Q3)	84.5 (63.0, 94.0)	92.0 (71.0, 107.0)	
Range (Min–Max)	43.0–140.0	55.0–132.0	

## Data Availability

The study present data from The PRINCESS randomized controlled Trial. The data are not publicly available in accordance with ethical approval and institutional regulations of patient data management.

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
