# Peer review of "Strategies of Advanced Airway Management in Out-of-Hospital Cardiac Arrest during Intra-Arrest Hypothermia: Insights from the PRINCESS Trial"

_jcm, 2022, doi:10.3390/jcm11216370_

Round 1

Reviewer 1 Report

Many previous studies have reported different results on the effects between airway management types applied to patients with pre-hospital cardiac arrest.
The reason why the results were different for each study suggests that there are other factors that influence the results as well as the timing of application of the airway management used in the pre-hospital stage.

However, in this study, there is an additional factor called intra cooling time that affects the results.

Therefore, problems may arise in the interpretation of research results.
This is because it is not known whether the difference between the two groups is due to the airway management method or the time until the initiation of intra-cooling.

The authors seem to have conducted the study assuming that the difference between the two groups was simply due to the difference in time until the onset of intra-cooling.

Therefore, attention is required to interpret the results of this study.

For a more accurate study, it is necessary to compare the prognosis changes according to the difference in the time required to start intra-cooling within the group using the same airway management method.

Author Response

Reviewer # 1

Many previous studies have reported different results on the effects between airway management types applied to patients with pre-hospital cardiac arrest.

The reason why the results were different for each study suggests that there are other factors that influence the results as well as the timing of application of the airway management used in the pre-hospital stage.

However, in this study, there is an additional factor called intra cooling time that affects the results.

Therefore, problems may arise in the interpretation of research results.

This is because it is not known whether the difference between the two groups is due to the airway management method or the time until the initiation of intra-cooling.

The authors seem to have conducted the study assuming that the difference between the two groups was simply due to the difference in time until the onset of intra-cooling.

Therefore, attention is required to interpret the results of this study.

For a more accurate study, it is necessary to compare the prognosis changes according to the difference in the time required to start intra-cooling within the group using the same airway management method.

We agree and we thank the reviewer for these very important comments and remark.

In this study, there may be several reasons why the EMS team started and also used a supraglottic airway device (SGA). The background is this: All different study sites within this study had a two tiered systems and the first EMS vehicle most usually used bag-mask ventilation or SGA before arrival of the second tier. In addition, in the majority of cases the intra-arrest cooling equipment used in this trial was carried out by the second tier. Thus, patients could have had the SGA placed by the crew from the first vehicle and then cooling was started after randomization and application of the transnasal evaporative cooling by the crew in the second tier. This may explain why the time to airway management differ between groups in advantage of SGA while the median time to cooling start is similar. This has now been clarified in the manuscript (intro, methods, and discussion).

Furthermore, as the reviewer point out, we acknowledge that the time until the initiation of intra-arrest cooling is one of several factors we believe may influence patient outcomes in this study. We have earlier studied this association in a previous publication in Intensive Care Medicine (Awad et al, 2020).

We have also as suggested now made additional analyses and have added a univariate logistic regression to examine the association between patient outcomes and the time until the initiation of intra-arrest cooling therapy or the time until successful airway device insertion, respectively. The outcome was used as a????the dependent variable and the time until airway device insertion or the time until the initiation of cooling were used as the independent variables. As seen in this analysis and fully in line with our previous finding, we found that the time until the initiation of intra-arrest cooling was associated with survival with good neurological outcome (defined as CPC 1-2, OR 0.96 [0.92 - 1.00], p= 0.045), full recovery (defined as CPC 1, 0.95 [0.90 - 0.99], p = 0.018= and overall survival (OR 0.95 [0.91 - 0.99], p = 0.017). These results are now displayed in the supplementary materials (supplemental table 2). These results might explain why we found no differences in outcomes between the two treatment groups, as there were in fact no statistically significant differences in the time until the initiation of intra-arrest cooling between the ETI- and SGA groups (As is shown in Figure 3B in the latest version of the manuscript).

The above cited data was derived from the analysis of the entire cohort, i.e. both ETI and SGA patients. However, we observed similar results regarding the relationship between patient outcomes and the time until intra-arrest cooling when examining the SGA- and ETI groups separately.  The only notable difference was that the confidence intervals were wider for the SGA group, which can likely be explained by the lower sample size (69 SGA patients and 259 ETI patients in the complete cohort before propensity score matching). For example, when examining the outcome of CPC 1-2, when assessing modification of treatment effect, we observed no significant interaction between airway device and the effect of time until intra-arrest cooling on patient outcomes (p interaction = 0.766). Thus, the relationship between the time until intra-arrest cooling and patient outcomes can be assumed to be independent of airway management. Therefore, we have decided to only display the results for the entire cohort in the manuscript, and not the data for each treatment group separately.

Reviewer 2 Report

Dear authors,

 I read the article Strategies of advanced airway management in out-of-hospital cardiac arrest during intra-arrest hypothermia: insights from the PRINCESS trial with great interest. The aim of the presented article (sub-study of the PRINCESS Trial) was to compare the effect on neurologic outcome among OHCA patients that had received airway management with endotracheal intubation (ETI) versus supraglottic airway (SGA) prior to trans-nasal evaporative intra-arrest cooling.

However, following questions should be addressed:

1. How was the time interval for successful airway management defined: from the start of CPR until successful airway management? Please describe it in the Methodology section.

2. Please add in the Table 1 following variables for compared groups according to the Utstein template:

- time interval to ROSC (affect also the results in Table 2)

- time interval to hospital admission

- No of patients with ROSC

- No of patients to hospital admission

Please also add in that table: - time to initiation of intranasal cooling

3. Regarding results in the Table 2 please describe in the Methodology section how were the patients in the presented study ventilated during CPR after successful airway management? Were they ventilated with the portable ventilators (what were the settings: tidal volume? frequency? PEEP, FiO2?, etc…) or with bag-valve connected to endotracheal tube or SGA? Were the settings changed after ROSC?

4. According to the presented results in Figure 3 I have several comments:

- Honestly, I am surprised with the median time needed for ETI (8 min!); in some cases, with up to 20 min?! Please describe which member of the team intubated the patient and what competencies he/she had for the intubation? According to that, please add in the Methodology section description of EMS included in the study: were they physician led or with only paramedics on board?

- Regarding the current guidelines at that time the intubation attempt should not take more than 30 sec and no more than 3 attempts should be done. After unsuccessful intubation an alternative approach should be undertaken. Why did the crew members insist for ETI so long? In which attempt did they successfully intubate the patient? How many patients were with ET misplacement? How was the correct ET placement confirmed? (etCO2)?

- I can assume (from practical experiences) that prolonged time interval for ETI lead to several chest compressions interruptions and therefore to less favorite outcome (also neurological)?!

- Did the prolong time interval for ETI also lead to the delayed initiation of intranasal cooling and therefore attenuates the beneficial effect of hypothermia? Could be that also the reason for less favorable neurological outcome? I believe that these afore-mentioned concerns should be addressed and also mentioned in the limitations section.

Best regards.

Author Response

Dear authors,

 I read the article Strategies of advanced airway management in out-of-hospital cardiac arrest during intra-arrest hypothermia: insights from the PRINCESS trial with great interest. The aim of the presented article (sub-study of the PRINCESS Trial) was to compare the effect on neurologic outcome among OHCA patients that had received airway management with endotracheal intubation (ETI) versus supraglottic airway (SGA) prior to trans-nasal evaporative intra-arrest cooling.

However, following questions should be addressed:

  1. How was the time interval for successful airway management defined: from the start of CPR until successful airway management? Please describe it in the Methodology section.

Thanks for this valuable comment. The time interval for successful airway management was defined as the time elapsed from the time of EMS arrival at the site of the arrest until the time of successful airway device placement. Similarly, we defined the time until the initiation of intra-arrest cooling as the time interval between EMS arrival and the initiation of trans-nasal evaporative cooling. This has now been clarified in the methodology section.

  1. Please add in the Table 1 following variables for compared groups according to the Utstein template:

- time interval to ROSC (affect also the results in Table 2)

- time interval to hospital admission

- No of patients with ROSC

- No of patients to hospital admission

Please also add in that table: - time to initiation of intranasal cooling

Point well taken. We have now added the requested parameters to Table 1, including the time until ROSC and the time of hospital arrival. The time to initiation of intranasal cooling is instead shown in Figure 3 B. Additionally, the number of patients who achieved sustained ROSC is presented in figure 2, as it was considered a secondary outcome of this study.

  1. Regarding results in the Table 2 please describe in the Methodology section how were the patients in the presented study ventilated during CPR after successful airway management? Were they ventilated with the portable ventilators (what were the settings: tidal volume? frequency? PEEP, FiO2?, etc…) or with bag-valve connected to endotracheal tube or SGA? Were the settings changed after ROSC?

This is indeed a very important issue. In the vast majority of cases, the patients were ventilated by bag-valve connected to the endotracheal tube or SGA in accordance to local practices and with the recommendation to ERC guidelines in regard to ventilation rate and oxygen level. In a limited number of patients (less than 10) from one study site, portable ventilators were used in some of the patients but we have no data in terms of tidal volumes, frequency or other variables. We have added this information in the method’s section.

  1. According to the presented results in Figure 3 I have several comments:

- Honestly, I am surprised with the median time needed for ETI (8 min!); in some cases, with up to 20 min?! Please describe which member of the team intubated the patient and what competencies he/she had for the intubation? According to that, please add in the Methodology section description of EMS included in the study: were they physician led or with only paramedics on board?

Point well taken and we do agree that this needs to be clarified to better understand the settings. Physicians or nurses trained in anesthesiology performed the ETI. The time delays were due to longer response times of the second tier as explained below. We have added a section in the Methods to explain this to the readers.

- Regarding the current guidelines at that time the intubation attempt should not take more than 30 sec and no more than 3 attempts should be done. After unsuccessful intubation an alternative approach should be undertaken. Why did the crew members insist for ETI so long? In which attempt did they successfully intubate the patient? How many patients were with ET misplacement? How was the correct ET placement confirmed? (etCO2)?

We thank the reviewer for these very important comments. In this study, there may be several reasons why the EMS team used a supraglottic airway device (SGA). All study sites had two tiered systems and the first vehicle used bag mask ventilation or SGA before arrival of the second tier. In addition, the intra-arrest cooling equipment was carried by the second tier. Thus, patients could have had the SGA placed by the crew from the first vehicle and then cooling was started after randomization and application of the transnasal evaporative cooling by the crew in the second tier. Among patients receiving bag mask ventilation, ETI was performed after arrival of the second tier. Thus, the time to ETI is calculated from the arrival time of the first unit until ETI is performed by the second unit. This may explain why the time to airway management differ between groups in advantage of SGA while the median time to cooling start is similar. This has now been clarified in the manuscript (intro, methods, and discussion). The use of SGA could also be due to the fact that ETI was difficult to perform in the field. However, we no data if this was the case.

The confirmation of the tube was done with etCO2 but this was only recorded in a limited number of patients.

- I can assume (from practical experiences) that prolonged time interval for ETI lead to several chest compressions interruptions and therefore to less favorite outcome (also neurological)?!

Thank you for the comment. As stated in the answer above, the main reason for the long time interval for ETI was due to the response time of the second tier and not by several attempts to intubate. We have performed additional analyses to study the association of time to airway and outcome and time to cooling initiation and outcome (se supplement). In line with previous findings we could see that the time to cooling is associated with good outcome while time to airway is not. We have clarified this in the methods and discussion.

We did not record data on the quality of cardiopulmonary resuscitation, such as information regarding chest compression interruptions. This is a limitation of this study.

- Did the prolong time interval for ETI also lead to the delayed initiation of intranasal cooling and therefore attenuates the beneficial effect of hypothermia? Could be that also the reason for less favorable neurological outcome? I believe that these afore-mentioned concerns should be addressed and also mentioned in the limitations section.

Thanks for very valuable comments. We found that there were no statistically significant differences between the the SGA- and ETI groups with regards to the time until the initiation of transnasal intra-arrest cooling, as is now shown in Figure 3B. In brief, we believe that this might primarily be due to logistical factors as the cooling equipment was carried by the second tier, see answer above.

Thus, although the patient's airway may have been secured using an SGA by the paramedic prior to the arrival of the pre-hospital emergency medicine physicians, intra-arrest cooling could only be initiated once the physician had arrived at the site of the arrest, making the time to hypothermia largely independent of the time until successful airway management

One might speculate that were these logistical issues to be resolved, then perhaps SGA could be superior to ETI, as SGA theoretically could allow for a reduction in the time until the initiation of transnasal evaporative cooling. This hypothesis is supported by the fact that the time until the initiation of intra-arrest cooling was significantly associated with patient outcomes, as can now be seen in the supplementary materials (Supplemental table 2). This hypothesis will have to be confirmed in future studies.

Best regards.

Round 2

Reviewer 1 Report

Since the intranasal cooling device could only be started after the arrival of the 2nd tier ambulance, the time to initiation of cooling and the outcome of cardiac arrest patients did not appear to be significantly affected by the airway management method. Although there are some limitations in study design and method, the authors appear to have reached appropriate conclusions. It is expected that the results of this study will be helpful in planning a pre-hospital intranasal cooling strategy in the future.

Author Response

Since the intranasal cooling device could only be started after the arrival of the 2nd tier ambulance, the time to initiation of cooling and the outcome of cardiac arrest patients did not appear to be significantly affected by the airway management method. Although there are some limitations in study design and method, the authors appear to have reached appropriate conclusions. It is expected that the results of this study will be helpful in planning a pre-hospital intranasal cooling strategy in the future.

Thank you for these comments and for the assistance to improve the manuscript

Reviewer 2 Report

Dear authors,

I found some following inconsistences (please address them) in the revised article Strategies of advanced airway management in out-of-hospital cardiac arrest during intra-arrest hypothermia: insights from the PRINCESS trial:

Abstract

Methods: “SGA was placed by paramedics in the first-tier ambulance while ETI was performed by physicians or anesthetic nurse in the second tier.« is not consistent with Methodology section in the article:  »All study sites had two-tiered EMS systems where the first vehicle used bask mask ventilation only or SGA with bag-valve ventilation connected to the SGA prior to arrival of the second tier. The second tier was manned by physicians or anesthetic nurses, trained in advance airway management including placing an SGA an endotracheal intubation.«

Results: “Median time to successful airway management from the arrival of the first-tier ambulance was 8 minutes for ETI and 4 minutes for SGA (p = 0.001).” ? According to the previous paragraph first tier ambulance use only bag mask ventilation or SGA not ETI?!

Conclusions: »These results imply that SGA could be used to shorten the time to transnasal evaporative cooling in OHCA.«   Data from the present article does not support this thesis. In Results section in the article: “However, we did not find any statistically significant difference between the SGA- and ETI groups regarding the time until the initiation of hypothermia treatment, which was on average 14 minutes in the ETI group (interquartile range 8-20) and 15 minutes in the SGA group (interquartile range 11 – 25), p = 0.053 (Figure 3B).«

Article

I have a question regarding (Outcome paragraph in Methods section): “The time until successful airway management was defined as the time interval which elapsed between EMS arrival at the site of the arrest and the time of successful airway device placement. «  Do you have any data which EMS vehicle (first- or second-tier) in the presented study was first on scene? Was it always EMS vehicle form first-tier? Please add this data in the Table 1. I find it essential for the data interpretation due to differences in competencies (SGA vs ETI) between first-tier and second-tier EMS.  If I understand it right, time elapsed from EMS arrival and successful ETI includes time interval between arrival of the first- and second-tier EMS vehicle and time interval for the ETI procedure? Therefore, it is not a surprise that SGA (which could be done by first-tier team) is placed faster?  Please add it in discussion/limitation section.

In Methods section (EMS paragraph) please correct used bag mask.

Best regards.

Author Response

Dear authors,

I found some following inconsistences (please address them) in the revised article Strategies of advanced airway management in out-of-hospital cardiac arrest during intra-arrest hypothermia: insights from the PRINCESS trial:

Abstract

Methods: “SGA was placed by paramedics in the first-tier ambulance while ETI was performed by physicians or anesthetic nurse in the second tier.« is not consistent with Methodology section in the article:  »All study sites had two-tiered EMS systems where the first vehicle used bask mask ventilation only or SGA with bag-valve ventilation connected to the SGA prior to arrival of the second tier. The second tier was manned by physicians or anesthetic nurses, trained in advance airway management including placing an SGA an endotracheal intubation.«

Results: “Median time to successful airway management from the arrival of the first-tier ambulance was 8 minutes for ETI and 4 minutes for SGA (p = 0.001).” ? According to the previous paragraph first tier ambulance use only bag mask ventilation or SGA not ETI?!

Conclusions: »These results imply that SGA could be used to shorten the time to transnasal evaporative cooling in OHCA.«   Data from the present article does not support this thesis. In Results section in the article: “However, we did not find any statistically significant difference between the SGA- and ETI groups regarding the time until the initiation of hypothermia treatment, which was on average 14 minutes in the ETI group (interquartile range 8-20) and 15 minutes in the SGA group (interquartile range 11 – 25), p = 0.053 (Figure 3B).«

Thank you for these valuable comments. We have now addressed these in accordance with your suggestions in the abstract.

Article

I have a question regarding (Outcome paragraph in Methods section): “The time until successful airway management was defined as the time interval which elapsed between EMS arrival at the site of the arrest and the time of successful airway device placement. «  Do you have any data which EMS vehicle (first- or second-tier) in the presented study was first on scene? Was it always EMS vehicle form first-tier? Please add this data in the Table 1. I find it essential for the data interpretation due to differences in competencies (SGA vs ETI) between first-tier and second-tier EMS.  If I understand it right, time elapsed from EMS arrival and successful ETI includes time interval between arrival of the first- and second-tier EMS vehicle and time interval for the ETI procedure? Therefore, it is not a surprise that SGA (which could be done by first-tier team) is placed faster?  Please add it in discussion/limitation section.

Thank you for this valuable comment. The differences in arrival times between EMS and ALS has now been described in a supplemental figure. As suggested, We have added this also in the text in the section of limitation.

In Methods section (EMS paragraph) please correct used bag mask.

This has now been corrected.

Best regards.